# Transfer Learning via $\ell_1$ Regularization

**Masaaki Takada**
Toshiba Corporation
Tokyo 105-0023, Japan
masaaki1.takada@toshiba.co.jp

**Hironori Fujisawa**
The Institute of Statistical Mathematics
Tokyo 190-8562, Japan
fujisawa@ism.ac.jp

## Abstract

Machine learning algorithms typically require abundant data under a stationary environment. However, environments are nonstationary in many real-world applications. Critical issues lie in how to effectively adapt models under an ever-changing environment. We propose a method for transferring knowledge from a source domain to a target domain via $\ell_1$ regularization in high dimension. We incorporate $\ell_1$ regularization of differences between source parameters and target parameters, in addition to an ordinary $\ell_1$ regularization. Hence, our method yields sparsity for both the estimates themselves and changes of the estimates. The proposed method has a tight estimation error bound under a stationary environment, and the estimate remains unchanged from the source estimate under small residuals. Moreover, the estimate is consistent with the underlying function, even when the source estimate is mistaken due to nonstationarity. Empirical results demonstrate that the proposed method effectively balances stability and plasticity.

## 1 Introduction

Machine learning algorithms typically require abundant data under a stationary environment. However, real-world environments are often nonstationary due to, for example, changes in the users' preferences, hardware or software faults affecting a cyber-physical system, or aging effects in sensors [39]. Concept drift, which means the underlying functions change over time, is recognized as a root cause of decreased effectiveness in data-driven information systems [25].

Under an ever-changing environment, critical issues lie in how to effectively adapt models to a new environment. Traditional approaches tried to detect concept drift based on hypothesis test [11, 26, 19, 5], but they are hard to capture continuously ever-changing environments. Continuously updating approaches, in contrast, are effective for complex concept drift by avoiding misdetection. These include tree-based methods [7, 16, 24] and ensemble-based methods [31, 20, 10]. Additionally, parameter-based transfer learning for transferring knowledge from past (source domains) to present (target domains) has been studied empirically and theoretically [27, 35, 21, 22]. They employed an empirical risk minimization with $\ell_2$ regularization, and the regularization was extended to strongly convex functions. However, these methods do not yield sparsity of parameter changes, so that even slight changes of data incur update of all parameters.

Specifically, we consider the problem of sparse regression [32, 14]. Sparse models are widely used in decision making since they have few active features and thus easy to obtain some insight. However, existing sparse regression methods are not necessarily effective for routine decision making, because they can significantly change parameters even when the data changes only slightly.

In this paper, we provide a method for transferring knowledge in high dimension via $\ell_1$ regularization that allows sparse estimates and changes. We incorporate the $\ell_1$ regularization of the difference between source parameters and target parameters into the ordinary Lasso regularization. The ordinary Lasso regularization has a role of restricting the model complexity in high dimension. The additional

$\ell_1$ regularization of difference plays a key role of sparse update, that is, only a small number of parameters are changed. Because of these two kinds of sparsity, it is easy to interpret and manage models and their changes. The proposed method has a single additional hyper-parameter compared to the ordinary Lasso. It controls the regularization strengths of estimates themselves and their changes, thereby balances stability and plasticity to mitigate so-called *stability-plasticity dilemma* [13, 6]. Therefore, our method transfers knowledge from past to present when the environment is stationary; while it discards the outdated knowledge when concept drift occurs.

Our main contribution is to give a method with clear theoretical justifications. We demonstrate three favorable characteristics for our method. First, our method presents a smaller estimation error than Lasso when the underlying functions do not change and the source estimate is the same as a target parameter. This indicates that our method effectively transfers knowledge under a stationary environment. Second, our method gives a consistent estimate even when the source estimate is mistaken, albeit with a weak convergence rate due to the phenomenon of so-called *negative transfer* [42]. This implies that our method can effectively discard the outdated knowledge and obtain new knowledge under nonstationary environment. Third, our method does not update estimates when the residuals of the predictions are small and the regularization is large. Hence, our method has an implicit stationarity detection mechanism.

The remainder of this paper is organized as follows. We begin with the description of the proposed method in Section 2. We also give some reviews on related work, including concept drift, transfer learning, and online learning. We next show some theoretical properties in Section 3. We finally illustrate empirical results in Section 4 and conclude in Section 5. All the proofs, as well as additional theoretical properties and empirical results, are given in the supplementary material.

## 2 Methods

### 2.1 Transfer Lasso

Let $X_i \in \mathcal{X}$ and $Y_i \in \mathbb{R}$ be the feature and response, respectively, for $i = 1, \ldots, n$. Consider a linear function

$$f_\beta(\cdot) = \sum_{j=1}^{p} \beta_j \psi_j(\cdot),$$

where $\beta = (\beta_j) \in \mathbb{R}^p$ and $\psi_j(\cdot)$ is a dictionary function from $\mathcal{X}$ to $\mathbb{R}$. Let the target function and noise be denoted by

$$f^*(\cdot) = f_{\beta^*}(\cdot) := \sum_{j=1}^{p} \beta_j^* \psi_j(\cdot) \text{ and } \varepsilon_i := Y_i - f^*(X_i),$$

and in matrix notion, $\mathbf{f}^* := \mathbf{X}\beta^*$ and $\varepsilon := \mathbf{y} - \mathbf{f}^*$, where $\mathbf{f}^* = (f^*(X_i)) \in \mathbb{R}^n$, $\mathbf{X} = (\psi_j(X_i)) \in \mathbb{R}^{n \times p}$, $\beta^* = (\beta_j^*) \in \mathbb{R}^p$, and $\mathbf{y} = (Y_i) \in \mathbb{R}^n$.

In high-dimensional settings, a reasonable approach to estimating $\beta^*$ is to assume sparsity of $\beta^*$, in which the cardinality $s = |S|$ of its support $S := \{j \in \{1, \ldots, p\} : \beta_j^* \neq 0\}$ satisfies $s \ll p$, and to solve the Lasso problem [32], given by

$$\min_{\beta \in \mathbb{R}^p} \left\{ \frac{1}{2n} \sum_{i=1}^{n} (Y_i - f_\beta(X_i))^2 + \lambda \|\beta\|_1 \right\}.$$

Lasso shrinks the estimate to zero and yields a sparse solution. We focus on the squared loss function, but it is applicable to other loss function, as seen in Section 4.3.

Suppose that we have an initial estimate of $\beta^*$ as $\tilde{\beta} \in \mathbb{R}^p$ and that the initial estimate is associated with the present estimate. Then, a natural assumption is that the difference between initial and present estimates is sparse. Thus, we employ the $\ell_1$ regularization of the estimate difference and incorporate it into the ordinary Lasso regularization as

$$\hat{\beta} = \operatorname*{argmin}_{\beta \in \mathbb{R}^p} \left\{ \frac{1}{2n} \sum_{i=1}^{n} (Y_i - f_\beta(X_i))^2 + \lambda \left( \alpha\|\beta\|_1 + (1-\alpha)\|\beta - \tilde{\beta}\|_1 \right) \right\} =: \mathcal{L}(\beta; \tilde{\beta}), \quad (1)$$

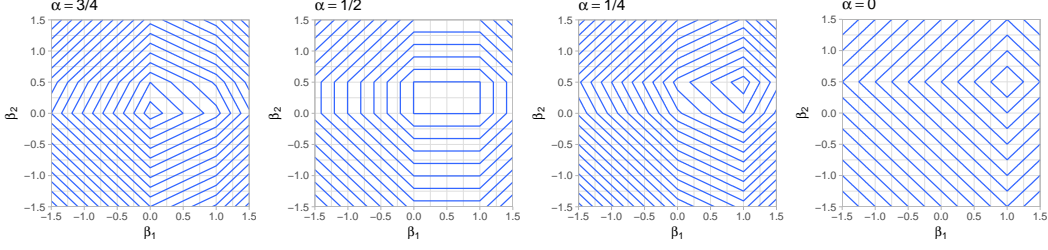

Figure 1: Contours of our regularizer for $\alpha = 3/4, 1/2, 1/4, 0$ with $\tilde{\beta} = (1, 1/2)^\top$.

where $\lambda = \lambda_n > 0$ and $0 \le \alpha \le 1$ are regularization parameters. We call this method "Transfer Lasso". There are two anchor points, zero and the initial estimate. The first regularization term in (1) shrinks the estimate to zero and induces the sparsity of the estimate. The second regularization term in (1) shrinks the estimate to the initial estimate and induces the sparsity of changes from the initial estimates. The parameter $\alpha$ controls the balance between transferring and discarding knowledge. It is preferable to transfer knowledge of the initial estimate when the underlying functions remain unchanged, while not preferable to transfer when a concept drift occurred. As a particular case, if $\alpha = 1$, Transfer Lasso reduces to ordinary Lasso and discards knowledge of the initial estimate. On the other hand, if $\alpha = 0$, Transfer Lasso reduces to Lasso predicting the residuals of the initial estimate, $\mathbf{y} - \mathbf{X}\tilde{\beta}$, and the initial estimate is utilized as a base learner. The regularization parameters, $\lambda$ and $\alpha$, are typically determined by cross validation.

Figure 1 shows the contours of our regularizer for $p = 2$. Contours are polygons pointed at $\beta_j = 0$ and $\beta_j = \tilde{\beta}_j$ so that our estimate can shrink to zero and the initial estimate. The regularization parameter $\alpha$ controls the shrinkage strengths to zero and the initial estimate. We also see that Transfer Lasso mitigates feature selection instability in the presence of highly correlated features. This is because the loss function tends to be parallel to $\beta_1 + \beta_2 = c$ for highly correlated features but the contours are not necessarily parallel to $\beta_1 + \beta_2 = c$ for a quadrant of $\tilde{\beta}$. For $\alpha = 1/2$, the sum of the two regularization terms equals $\lambda\tilde{\beta}$ in the rectangle of $\beta_j \in [\min\{\tilde{\beta}_j, 0\}, \max\{\tilde{\beta}_j, 0\}]$. If the least square estimate lies in this region, it becomes the solution of Transfer Lasso, that is, it does not have any estimation bias.

## 2.2 Algorithm and Soft-Threshold Function

We provide a coordinate descent algorithm for Transfer Lasso. It is guaranteed to converge to a global optimal solution [36], because the problem is convex and the penalty is separable. Let $\beta$ be the current value. Consider a new value $\beta_j$ as a minimizer of $\mathcal{L}(\beta; \tilde{\beta})$ when other elements of $\beta$ except for $\beta_j$ are fixed. We have

$$\partial_{\beta_j}\mathcal{L}(\beta; \tilde{\beta}) = -\frac{1}{n}\mathbf{X}_j^\top(\mathbf{y} - \mathbf{X}_{-j}\beta_{-j}) + \beta_j + \lambda\alpha\,\mathrm{sgn}(\beta_j) + \lambda(1-\alpha)\,\mathrm{sgn}(\beta_j - \tilde{\beta}_j) = 0,$$

where $\mathbf{X}_j$ and $\mathbf{X}_{-j}$ denote the $j$-th column of $X$ and $X$ without $j$-th column, respectively, and $\mathrm{sgn}(\cdot)$ denotes the sign function, hence we obtain the update rule as

$$\beta_j \leftarrow \mathcal{T}\left(\frac{1}{n}\mathbf{X}_j^\top(\mathbf{y} - \mathbf{X}_{-j}\beta_{-j}), \lambda, \lambda(2\alpha - 1), \tilde{\beta}_j\right),$$

where

$$\mathcal{T}(z, \gamma_1, \gamma_2, b) := \begin{cases} 0 & \text{for} & -\gamma_1 \le z \le \gamma_2 & | & -\gamma_2 \le z \le \gamma_1 \\ b & \text{for} & \gamma_2 + b \le z \le \gamma_1 + b & | & -\gamma_1 + b \le z \le -\gamma_2 + b \\ z - \gamma_2\,\mathrm{sgn}(b) & \text{for} & \gamma_2 \le z \le \gamma_2 + b & | & -\gamma_2 + b \le z \le -\gamma_2 \\ z - \gamma_1\,\mathrm{sgn}(z) & \text{for} & \text{otherwise} & | & \text{otherwise.} \end{cases}$$

$$\begin{array}{cc} b \ge 0 & b \le 0 \end{array}$$

The computational complexity of Transfer Lasso is the same as ordinary Lasso.

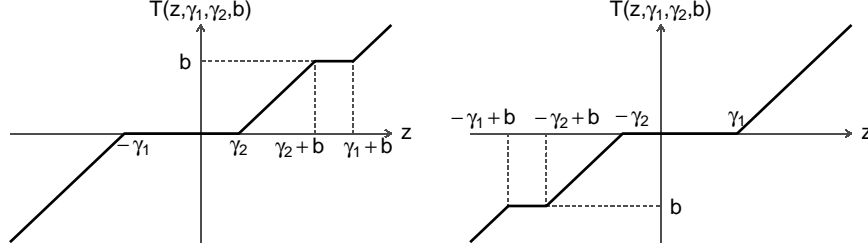

Figure 2: The soft-thresholding function for $b \geq 0$ (left) and $b \leq 0$ (right) with $\gamma_1 > 0$ and $|\gamma_2| \leq \gamma_1$.

Figure 2 shows the soft-threshold function $\mathcal{T}(z, \gamma_1, \gamma_2, b)$ with $|\gamma_2| = |\lambda(2\alpha - 1)| \leq \lambda = \gamma_1$. There are two steps at 0 and $b = \tilde{\beta}$. This implies that each parameter $\hat{\beta}_j$ is likely to be zero or the initial estimate $\tilde{\beta}_j$. As $\alpha$ approaches 1, the step of the initial estimate disappears and reduces to the standard soft-thresholding function. As $\alpha$ instead approaches 0, the step of zero disappears and the parameter only shrinks to the initial estimate.

## 2.3 Related Work

Transfer Lasso relates to concept drift, transfer learning, and online learning, as reviewed below.

Concept drift is a scenario where underlying functions change over time [12, 6]. There are two strategies for learning concept drift, active and passive approaches. The active approach explicitly detects concept drift and updates the model [11, 26, 19, 5]. Although they work well for abrupt concept drift, it is hard to capture gradual concept drift. The passive approach, on the other hand, continuously updates the model every time. There are some ensemble learner methods [31, 20, 10] and single learner methods including tree-based models [7, 16, 24] and neural network-based models [41, 3, 4]. They are effective for gradual and complex concept drift empirically. However, most methods always update models even if an environment is stationary, and these ad-hoc algorithms are hard to support their effectiveness theoretically. In contrast, Transfer Lasso can remain the estimate unchanged when the underlying functions do not change and also has some theoretical justifications.

Transfer learning is a framework that improves the performance of learners on target domains by transferring the knowledge of source domains [28, 38, 42]. This paper considers a homogeneous inductive transfer learning setting, which means that feature spaces and label spaces are the same between the source and target domains, and the label information of both domains is available. Hypothesis transfer learning is a typical approach for this problem [27, 35, 21, 22]. It transfer knowledge of source stimate $\tilde{\beta}$ by solving

$$\operatorname*{argmin}_{\beta \in \mathbb{R}^p} \frac{1}{2n} \sum_{i=1}^{n} (Y_i - f_\beta(X_i))^2 + \lambda \|\beta - \tilde{\beta}\|_2^2.$$

Similarly, single-model knowledge transfer [34] and multi-model knowledge transfer [33] employed another regularization $\|\beta - \alpha\tilde{\beta}\|_2^2 = (1-\alpha)\|\beta\|_2^2 + \alpha\|\beta - \tilde{\beta}\|_2^2 + \text{const}$, where $\alpha$ is a hyper-parameter. The $\ell_2$ regularization and its extension of strongly convex regularization are easy to analyze the generalization ability theoretically. In contrast, Transfer Lasso employs $\ell_1$ regularization, so that it yields sparsity of the changes of the estimates and requires different techniques for theoretical analysis. Sparsity is beneficial in practice because we can interpret and manage models by handling only a small number of estimates and their changes.

Online learning is a method where a learner attempts to learn from a sequence of instances one-by-one at each time [15]. The algorithms consist of the minimization of a cost function including $\|\beta - \beta^t\|_2^2$, where $\beta^t$ is a previous estimate, to stabilize the optimization [23, 9, 40, 8]. This is related to Transfer Lasso by regarding $\beta^t$ as an initial estimate, although these online algorithms work under a stationary environment.

## 3 Theoretical Properties

We analyze the statistical properties of Transfer Lasso. First, we construct estimation error bound and demonstrate the effectiveness under the correct and incorrect initial estimate. Second, we explicitly derive the condition that the model remains unchanged. Third, we investigate the behavior of Transfer Lasso when an initial estimate is a Lasso solution using another dataset.

### 3.1 Estimation Error

We prepare the following assumption and definition for our analysis.

**Assumption 1** (Sub-Gaussian). *The noise sequence $\{\varepsilon_i\}_{i=1}^n$ is i.i.d. sub-Gaussian with $\sigma$, i.e.,*

$$\mathrm{E}[\exp(t\varepsilon)] \le \exp\left(\frac{\sigma^2 t^2}{2}\right), \quad \forall t \in \mathbb{R}.$$

**Definition 1** (Generalized Restricted Eigenvalue Condition (GRE)). *We say that the generalized restricted eigenvalue condition holds for a set $\mathcal{B} \subset \mathbb{R}^p$ if we have*

$$\phi = \phi(\mathcal{B}) := \inf_{v \in \mathcal{B}} \frac{v^\top \frac{1}{n} \mathbf{X}^\top \mathbf{X} v}{\|v\|_2^2} > 0.$$

The GRE condition is a generalized notion of the restricted eigenvalue condition [1, 2, 14]. From the above assumption and definition, we have the following theorems and corollary. Let $\Delta := \tilde{\beta} - \beta^*$ and $v_S$ be the vector $v$ restricted to the index set $S$.

**Theorem 1** (Estimation Error). *Suppose that Assumption 1 is satisfied. Suppose that the generalized restricted eigenvalue condition (Definition 1) holds for $\mathcal{B} = \mathcal{B}(\alpha, c, \Delta)$, where*

$$\mathcal{B}(\alpha, c, \Delta) := \{v \in \mathbb{R}^p : (\alpha - c)\|v_{S^c}\|_1 + (1 - \alpha)\|v - \Delta\|_1 \le (\alpha + c)\|v_S\|_1 + (1 - \alpha)\|\Delta\|_1\},$$

*with some constant $c > 0$. Then, we have*

$$\|\hat{\beta} - \beta^*\|_2^2 \le \frac{(\alpha + c)^2 \lambda_n^2 s}{\phi^2} \left(1 + \sqrt{1 + \frac{2(1 - \alpha)\phi\|\Delta\|_1}{(\alpha + c)^2 \lambda_n s}}\right)^2 \tag{2}$$

*with probability at least $1 - \nu_{n,c}$, where $\nu_{n,c} := \exp(-nc^2\lambda_n^2/2\sigma^2 + \log(2p))$.*

The estimation error bound for Lasso is obtained from (2) with $\alpha = 1$ and $\Delta = 0$ as $4(1 + c)^2\lambda_n^2 s/\phi(\mathcal{B}_0)^2$, where $\mathcal{B}_0 = \mathcal{B}(1, c, 0) = \{v \in \mathbb{R}^p : (1 - c)\|v_{S^c}\|_1 \le (1 + c)\|v_S\|_1\}$. Consider the case $\tilde{\beta} = \beta^*$, that is, $\Delta = 0$. Then, the estimation error bound of Transfer Lasso reduces to $4(\alpha + c)^2\lambda_n^2 s/\phi(\mathcal{B}(\alpha, c, 0))^2$, where $\mathcal{B}(\alpha, c, 0) = \{v \in \mathbb{R}^p : (1 - c)\|v_{S^c}\|_1 \le (2\alpha - 1 + c)\|v_S\|_1\}$. Because $\mathcal{B}(\alpha, c, 0) \subset \mathcal{B}_0$ and so $\phi(\mathcal{B}(\alpha, c, 0)) \ge \phi(\mathcal{B}_0)$, the bound of Transfer Lasso ($\alpha < 1$) is smaller than that of Lasso.

The GRE condition for $\mathcal{B}(\alpha, c, \Delta)$ is not so restricted. The ordinary restricted eigenvalue condition holds for quite general classes of Gaussian matrices with high probability [29]. The GRE condition for $\mathcal{B}(\alpha, c, \Delta)$ with $\Delta = 0$ holds under a milder condition because $\mathcal{B}(\alpha, c, \Delta) \subset \mathcal{B}_0$. The GRE condition for $\mathcal{B}(\alpha, c, \Delta)$ with $2\alpha - c - 1 > 0$ holds as well except for constant factors because $\mathcal{B}(\alpha, c, \Delta) \subset \{v \in \mathbb{R}^p : (2\alpha - c - 1)\|v_{S^c}\|_1 \le (1 + c)\|v_S\|_1\}$.

**Theorem 2** (Convergence Rate). *Assume the same conditions as in Theorem 1. Then, with probability at least $1 - \nu_{n,c}$, we have*

$$\|\hat{\beta} - \beta^*\|_2^2 = O\left((\alpha + c)^2\lambda_n^2 s + (1 - \alpha)\lambda_n\|\Delta\|_1\right), \quad as \ \lambda_n \to 0.$$

Let $\lambda_n = O(\sqrt{\log p/n})$ and $\|\Delta\|_1 = O(s\sqrt{\log p/n})$. The order of $\lambda_n$ comes from the constant value of $\nu_{n,c}$, and the order of $\Delta$ is as in the ordinary Lasso rate. Then, the convergence rate is evaluated as $\|\hat{\beta} - \beta^*\|_2^2 = O(s \log p/n)$, which is an almost minimax optimal [30].

Let us consider a misspecified initial estimate, $\|\Delta\|_1 \nrightarrow 0$. For example, the case $\|\Delta\|_1 = O(s)$ is obtained when the initial estimate $\tilde{\beta}$ fails to detect the true value $\beta^*$, but most of the zeros are truly identified. Transfer Lasso estimates retain consistency even in this situation when $\|\Delta\|_1\lambda_n \to 0$, although the convergence rate becomes worse as $\|\hat{\beta} - \beta^*\|_2^2 = O(\|\Delta\|_1\sqrt{\log p/n})$ if $\alpha < 1$. This implies that negative transfer can happen but not severely, and is avoidable by setting $\alpha = 1$.

**Theorem 3** (Feature Screening)**.** *Assume the same conditions as in Theorem 1. Suppose that the beta-min condition*

$$|\beta_S^*| > \frac{(\alpha + c)^2 \lambda_n^2 s}{\phi^2} \left( 1 + \sqrt{1 + \frac{2(1-\alpha)\phi\|\Delta\|_1}{(\alpha + c)^2 \lambda_n s}} \right)^2$$

*is satisfied. Then, we have $S \subset \mathrm{supp}(\hat\beta)$ with probability at least $1 - \nu_{n,c}$.*

This theorem implies that Transfer Lasso succeeds in feature screening if the true parameters are not so small. The minimum value of true parameters for Transfer Lasso can be smaller than that for Lasso when $\|\Delta\|_1$ is small.

### 3.2 Unchanging Condition

The next theorem shows that the estimate remains unchanged under a certain condition.

**Theorem 4** (Unchanging Condition)**.** *Let $r(\beta) := \mathbf{y} - \mathbf{X}\beta$. There exists an unchanging solution $\hat\beta = \tilde\beta$ if and only if*

$$\left| \frac{1}{n} \mathbf{X}_j^\top r(\tilde\beta) \right| \le \lambda \ \text{ for } \ \forall j \ \text{ s.t. } \ \tilde\beta_j = 0, \ \text{ and}$$

$$- \lambda \left( (1-\alpha) - \alpha \,\mathrm{sgn}(\tilde\beta_j) \right) \le \frac{1}{n} \mathbf{X}_j^\top r(\tilde\beta) \le \lambda \left( (1-\alpha) + \alpha \,\mathrm{sgn}(\tilde\beta_j) \right) \ \text{ for } \ \forall j \ \text{ s.t. } \ \tilde\beta_j \ne 0.$$

*In addition, there exists a zero solution $\hat\beta = 0$ if and only if*

$$\left| \frac{1}{n} \mathbf{X}_j^\top r(0) \right| \le \lambda \ \text{ for } \ \forall j \ \text{ s.t. } \ \tilde\beta_j = 0, \ \text{ and}$$

$$- \lambda \left( \alpha + (1-\alpha) \,\mathrm{sgn}(\tilde\beta_j) \right) \le \frac{1}{n} \mathbf{X}_j^\top r(0) \le \lambda \left( \alpha - (1-\alpha) \,\mathrm{sgn}(\tilde\beta_j) \right) \ \text{ for } \ \forall j \ \text{ s.t. } \ \tilde\beta_j \ne 0.$$

This theorem shows that the estimate remains unchanged if and only if correlations between residuals and features are small and $\lambda$ is large. This is useful for constructing a search space for $\lambda$. We determine a sequence of $\lambda$ from $\lambda_{\max}$ to $\lambda_{\min}$, where $\lambda_{\max}$ is the smallest value for which all coefficients are zero or initial estimates by Theorem 4, and $\lambda_{\min}$ is defined by a user specified fraction of $\lambda_{\min}/\lambda_{\max}$.

### 3.3 Transfer Lasso as a Two-Stage Estimation

The initial estimate $\tilde\beta$ is arbitrary. We investigate the behavior of Transfer Lasso as a two-stage estimation. We suppose that the initial estimate is a Lasso solution using another dataset $\mathbf{X}' \in \mathbb{R}^{m \times p}$, $\mathbf{y}' \in \mathbb{R}^m$, and the true parameter $\tilde\beta^*$. Define $S' := \mathrm{supp}(\tilde\beta^*)$, $s' := |S'|$, and $\Delta^* := \tilde\beta^* - \beta^*$. Then, we have the following corollary.

**Corollary 5.** *Suppose that Assumption 1 is satisfied and the generalized restricted eigenvalue condition (Definition 1) holds with $\phi' = \phi'(\mathcal{B}')$ and $\mathcal{B}' = \mathcal{B}'(1, c', 0)$ on $\mathbf{X}', \mathbf{y}'$, and $\tilde\beta^*$. Assume the same conditions as in Theorem 1. Then, with probability at least $1 - \nu_{n,c} - \nu_{m,c'}$, we have*

$$\|\hat\beta - \beta^*\|_2^2 \le \frac{(\alpha + c)^2 \lambda_n^2 s}{\phi^2} \left( 1 + \sqrt{1 + \frac{4(1-\alpha)(1+c')\phi\lambda_m s'}{(\alpha + c)^2 \phi' \lambda_n s} + \frac{2(1-\alpha)\phi\|\Delta^*\|_1}{(\alpha + c)^2 \lambda_n s}} \right)^2 .$$

If there are abundant source data but few target data ($m \gg n$ and $\lambda_m \ll \lambda_n$), and the same true parameters ($\Delta^* = 0$), then we have $\|\hat\beta - \beta^*\|_2^2 \lesssim 4(\alpha + c)^2 \lambda_n^2 s/\phi^2$. This implies that Transfer Lasso with a small $\alpha$ is beneficial in terms of the estimation error. Additionally, we can see a similar weak convergence rate as in Theorem 2 even when $\|\Delta^*\|_1 \nrightarrow 0$.

## 4 Empirical Results

We first present two numerical simulations in concept drift and transfer learning scenarios. We then show real-data analysis results.

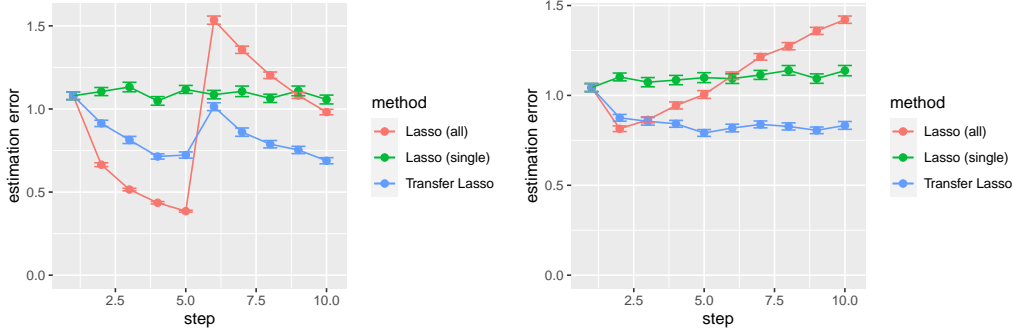

Figure 3: Estimation errors under the scenarios I (left; abrupt concept drift) and II (right; gradual concept drift).

## 4.1 Concept Drift Simulation

We first simulated concept drift scenarios. We used ten datasets $\{(\mathbf{X}^{(k)};\mathbf{y}^{(k)})\}_{k=1}^{10}$, and each dataset is generated by a linear regression $\mathbf{y}^{(k)} = \mathbf{X}^{(k)}\beta^{(k)} + \varepsilon^{(k)}$, where $\mathbf{y}^{(k)} \in \mathbb{R}^n$, $\mathbf{X}^{(k)} \in \mathbb{R}^{n \times p}$, $\beta^{(k)} \in \mathbb{R}^p$, $\varepsilon^{(k)} \in \mathbb{R}^n$, $n = 50$, $p = 100$, and $s = 10$. Elements of $\mathbf{X}^{(k)}$ and $\varepsilon^{(k)}$ were randomly generated from a standard Gaussian distribution. We examined two nonstationary scenarios, abrupt concept drift and gradual concept drift. Following these scenarios, we arranged different parameter sequences $\{\beta^{(k)}\}_{k=1}^{10}$.

Scenario I (Abrupt concept drift scenario). The underlying model suddenly changes drastically. At step $k = 1$, ten active features are randomly selected, and their coefficients are randomly generated from a uniform distribution of $[-1, 1]$. The former steps ($k = 1, \ldots, 5$) use the same $\beta$. At step $k = 6$, five active features are abruptly switched to other features, and their coefficients are also assigned in the same way. The remaining steps ($k = 6, \ldots, 10$) use the same values as $k = 6$.

Scenario II (Gradual concept drift scenario). The underlying model gradually changes. The first step is the same as in Scenario I. Then, at every step, one active feature switches to another, with its coefficient assigned from a uniform distribution.

We compared three methods, including our proposed method. (i) Lasso (all): We built the $k$-th model by Lasso using the first through $k$-th datasets. (ii) Lasso (single): We built the $k$-th model by Lasso using only a single $k$-th dataset. (iii) Transfer Lasso: We sequentially built each model by Transfer Lasso. For the $k$-th model, we applied Transfer Lasso to the $k$-th dataset, along with an initial estimate using Transfer Lasso applied to the $(k-1)$-th dataset. We used Lasso for the first model.

The regularization parameters $\lambda$ and $\alpha$ were determined by ten-fold cross validation. The parameter $\lambda$ was selected by a decreasing sequence from $\lambda_{\max}$ to $\lambda_{\max} * 10^{-4}$ in log-scale, where $\lambda_{\max}$ was calculated as in Section 3.2. The parameter $\alpha$ was selected among $\{0, 0.25, 0.5, 0.75, 1\}$. Each dataset was centered and standardized such that $\bar{\mathbf{y}} = 0$, $\bar{\mathbf{X}}_j = 0$ and $\text{sd}(\mathbf{X}_j) = 1$ in preprocessing.

Figure 3 shows the $\ell_2$-error for estimated parameters at each step. Averages and standard errors for the $\ell_2$-errors were evaluated in 100 experiments. In Scenario I, although Lasso (all) outperformed the others when the environment was stationary, it incurred significant errors after the abrupt concept drift. In contrast, Transfer Lasso gradually reduced estimation errors as the steps proceeded, and was not so worse when abrupt the concept drift occurred. Transfer Lasso always outperformed Lasso (single). In Scenario II, Transfer Lasso outperformed the others at most steps so that it balanced transferring and discarding knowledge. Lasso (all) used enough instances but induced a large estimation bias because various concepts (true models) exist in the datasets. Lasso (single) might not induce estimation bias, but incurred a lack of instances due to using only a single dataset.

## 4.2 Transfer Learning Simulation

We simulated a transfer learning scenario in which there were abundant source data but few target data. We used $\mathbf{y}^s = \mathbf{X}^s\beta^s + \varepsilon$ and $\mathbf{y}^t = \mathbf{X}^t\beta^t + \varepsilon$ for a source and target domain, respectively,

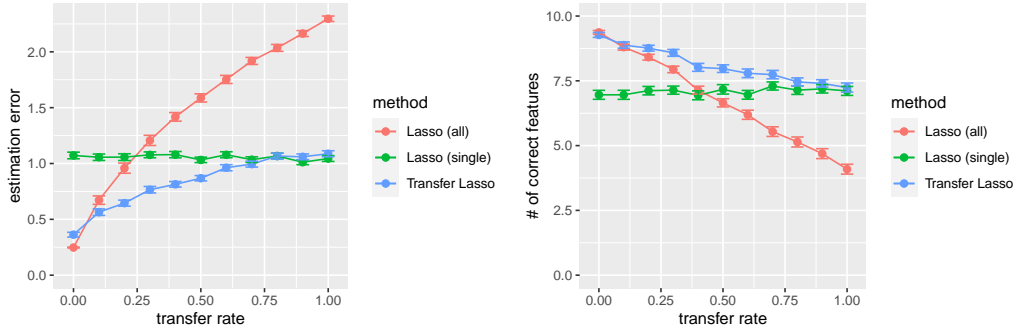

Figure 4: Estimation errors (left) and number of correct selected features (right) for transfer learning simulations.

where $\mathbf{X}^s \in \mathbb{R}^{n_s \times p}$, $\mathbf{X}^t \in \mathbb{R}^{n_t \times p}$, $n_s = 500$, $n_t = 50$, and $p = 100$. In the source domain, we generated $\beta^s$ in the same manner as in the concept drift simulation. For $\beta^t$ in the target domain, we switched each active features in $\beta^s$ to another feature at a "transfer rate" probability of 0 to 1. We compared three methods: Lasso (all), Lasso (single), and Transfer Lasso. Regularization parameters were determined in the same manner as above.

Figure 4 shows the results of the transfer learning simulations. Averages and standard errors for the $\ell_2$-errors were evaluated in 100 experiments. Transfer Lasso outperformed others in terms of $\ell_2$-error at almost all transfer rates, although Lasso (all) dominated when the transfer rate was zero, and Lasso (single) slightly dominated when the transfer rates were high. Transfer Lasso also showed the best accuracy in terms of feature screening.

## 4.3 Newsgroup Message Data

The newsgroup message data[1] comprises messages from Usenet posts on different topics. We basically followed the concept drift experiments in [18] and used preprocessed data[2]. The problem is to predict either the user is interested in email messages or not. There are 1500 examples and 913 attributes of boolean bag-of-words features. We suppose that the user is interested in the topics of space and baseball in the first 600 examples, while the user's interest changes to the topic of medicine in the remaining 900 examples. Thus, there is a abrupt concept drift of user's interests. The examples were divided into 30 batches without changing the order of the samples, each containing 50 examples. We trained models using each batch and evaluated them using the next batch. We compared three methods: Lasso (all), Lasso (single), and Transfer Lasso. Since this is a classification problem, we changed the squared loss function in (1) to the logistic loss. We used the coordinate descent algorithms as well. Regularization parameters were determined by ten-fold cross validation in the same manner as above except for $\alpha = 0.501$ instead of $\alpha = 0.5$ because of computational instability for binary features.

Figure 5 shows the results. Transfer Lasso outperformed Lasso (single) at almost all steps in terms of AUC (area under the curve). Lasso (all) performed well before concept drift (until 12-th batch), but significantly worsened after drift (from 13-th batch). Transfer Lasso showed stable behaviors of the estimates, and some coefficients remained unchanged. These results indicate that Transfer Lasso can follow data tendencies with minimal changes in the model.

## 5 Conclusion

We proposed and analyzed the $\ell_1$ regularization-based transfer learning framework. This approach is applicable to any parametric models, including GLM, GAM, and neural networks.

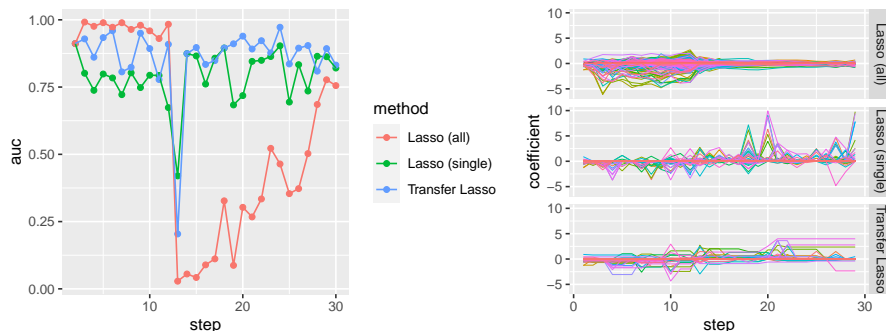

Figure 5: AUC (left) and coefficients (right) for newsgroup message data. Each coefficient is colored for legibility.

## Broader Impact

In this paper, sparsity meets transfer learning.

Sparsity has a key role in model transparency because a sparse model explains a phenomenon with few parameters. This is why Lasso is widely used in science such as genomics and economics, and in industries such as advanced electronics/semiconductors, chemicals, and health-care systems.

Our motivating examples of sparse estimation includes quality management in manufacturing. Production yield is one of the primary interests in factories and plants. Using Lasso, quality managers can screen and identify important factors for the yield from thousands or millions of candidates.

Here, we describe five types of impact of our approach along with possible applications, although our approach might have many potential impact/applications.

First, our approach enhances the efficiency of routine decision making. In manufacturing applications, quality managers need to analyze the factors behind yield fluctuations on a daily, weekly, or monthly basis. Since our approach can highlight the changes of parameters, they have only to check the difference from the past, which greatly streamlines the analysis and decision making.

Second, our approach also enhances the manageability of many models. In manufacturing applications, there are many kinds of products, so that many models are necessary. By transferring models from base products (source parameters) to the derivative products (target parameters), the total amount of active parameters is reduced, making it easier to manage a large number of models.

Third, our approach improves the model accuracy and robustness for high-dimensional small-sample data. Data scientists would easily build models from insufficient data, and furthermore, these modeling could be (semi-)automated.

Fourth, transferring knowledge among different companies is another possible application. Our approach only shares model parameters instead of data itself, hence secure and privacy-preserving transfer learning is possible.

Finally, one negative perspective could be transferring wrong knowledge, resulting in inaccurate models and hence incorrect/biased knowledge. However, we can decide whether the prior knowledge is transferred or not by controlling hyper-parameters. Additionally, we can incorporate our domain knowledge into the initial estimate. For example, a non-zero value of a certain initial estimate can be set to zero, or it can be replaced by another highly correlated feature. Therefore, we believe that this kind of concern could be overcome using appropriate domain knowledge. Such collaboration between human knowledge and real-world data is a key to model-based decision making, and it leads to a new paradigm of theory-guided data science [17] and informed machine learning [37].

## Acknowledgments

The authors received no third party funding for this work.

## Footnotes

[1]https://kdd.ics.uci.edu/databases/20newsgroups/20newsgroups.html

[2]http://lpis.csd.auth.gr/mlkd/concept_drift.html

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
