[Supplementary Material]

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

# A  Additional Theoretical Properties

We provide additional theoretical properties.

## A.1  Sign Recovery of Transfer Lasso

**Theorem 6** (Sign Recovery). *Assume that $\mathbf{X}_S^\top \mathbf{X}_S/n = I$. Then, we have $\operatorname{sgn}(\hat{\beta}) = \operatorname{sgn}(\beta^*)$ if and only if*

$$\operatorname{sgn}\left(\beta_S^* - w_S\right) = \operatorname{sgn}(\beta_S^*), \tag{3}$$

$$\left|\frac{1}{n}\mathbf{X}_{S^c}^\top \mathbf{X}_S w_S + \frac{1}{n}\mathbf{X}_{S^c}^\top \varepsilon + (1-\alpha)\lambda\operatorname{sgn}(\tilde{\beta}_{S^c})\right| \le \lambda\left(\alpha + (1-\alpha)1\{\tilde{\beta}_{S^c} = 0\}\right), \tag{4}$$

*where*

$$w_j := \begin{cases}
(2\alpha-1)\lambda\operatorname{sgn}(\beta_j^*) - \frac{1}{n}\mathbf{X}_j^\top \varepsilon, & \text{for } \beta_j^* > 0 \text{ and } \frac{1}{n}\mathbf{X}_j^\top \varepsilon - \Delta_j < \lambda(2\alpha-1), \\
& \text{for } \beta_j^* < 0 \text{ and } \frac{1}{n}\mathbf{X}_j^\top \varepsilon - \Delta_j > -(2\alpha-1)\lambda, \\
-\Delta_j, & \text{for } \beta_j^* > 0 \text{ and } (2\alpha-1)\lambda \le \frac{1}{n}\mathbf{X}_j^\top \varepsilon - \Delta_j \le \lambda, \\
& \text{for } \beta_j^* < 0 \text{ and } -\lambda \le \frac{1}{n}\mathbf{X}_j^\top \varepsilon - \Delta_j \le -(2\alpha-1)\lambda, \\
\lambda\operatorname{sgn}(\beta_j^*) - \frac{1}{n}\mathbf{X}_j^\top \varepsilon, & \text{for } \beta_j^* > 0 \text{ and } \frac{1}{n}\mathbf{X}_j^\top \varepsilon - \Delta_j > \lambda, \\
& \text{for } \beta_j^* < 0 \text{ and } \frac{1}{n}\mathbf{X}_j^\top \varepsilon - \Delta_j < -\lambda.
\end{cases}$$

**Remark 1.** *If $\varepsilon = 0$ and $\tilde{\beta}_{S^c} = 0$, the condition reduces to*

$$\operatorname{sgn}\left(\beta_S^* - w_S\right) = \operatorname{sgn}(\beta_S^*),$$

$$\left|\frac{1}{n}\mathbf{X}_{S^c}^\top \mathbf{X}_S w_S\right| \le \lambda,$$

*where*

$$w_j := \begin{cases}
0, & \text{for } \left(\Delta_j \ge 0 \text{ if } \beta_j^* > 0\right) \text{ or } \left(\Delta_j \le 0 \text{ if } \beta_j^* < 0\right) \\
-\Delta_j, & \text{for } \left(-\lambda \le \Delta_j \le 0 \text{ if } \beta_j^* > 0\right) \text{ or } \left(0 \le \Delta_j \le \lambda \text{ if } \beta_j^* < 0\right) \\
\lambda\operatorname{sgn}(\beta_j^*), & \text{for } \left(\Delta_j \le -\lambda \text{ if } \beta_j^* > 0\right) \text{ or } \left(\Delta_j \ge \lambda \text{ if } \beta_j^* < 0\right).
\end{cases}$$

*For the ordinary Lasso with $\varepsilon = 0$, the condition reduces to*

$$\operatorname{sgn}\left(\beta_S^* - \lambda\operatorname{sgn}(\beta_S^*)\right) = \operatorname{sgn}\left(\beta_S^*\right),$$

$$\left|\frac{1}{n}\mathbf{X}_{S^c}^\top \mathbf{X}_S \operatorname{sgn}(\beta_S^*)\right| \le 1.$$

*Since it holds that $|w_S| \le \lambda$, the condition of the Transfer Lasso is milder than that of the ordinary Lasso.*

**Theorem 7** (Sign Recovery under Sub-Gaussian Noise). *Suppose that Assumption 1 is satisfied. Suppose that the generalized restricted eigenvalue condition (Definition 1) holds for $\mathcal{B} = \mathcal{B}(\alpha, c, \Delta)$ Assume that $\mathbf{X}_S^\top \mathbf{X}_S/n = I$. Then, we have $\operatorname{sgn}(\hat{\beta}) = \operatorname{sgn}(\beta^*)$ if*

$$|\Delta_S| \le \frac{1}{2}\lambda_n,$$

$$\beta_{\min}^* > \lambda_n \max\left\{\frac{3}{2} - 2\alpha, 2\alpha - \frac{1}{2}\right\},$$

$$\tilde{\beta}_{S^c} = 0,$$

$$\left\|\frac{1}{n}\mathbf{X}_S^\top \mathbf{X}_{S^c}\right\|_\infty \le 1.$$

*with probability at least $1 - \exp(-n\lambda_n^2/8\sigma^2 + \log(2p))$.*

## A.2 Sign Unchanging Condition

**Theorem 8** (Sign Unchanging Condition). *Assume that $\mathbf{X}_S^\top \mathbf{X}_S / n = I$. Then, we have $\mathrm{sgn}(\hat{\beta}) = \mathrm{sgn}(\tilde{\beta})$ if and only if*

$$\mathrm{sgn}\left(\tilde{\beta}_{\tilde{S}} - w_{\tilde{S}}\right) = \mathrm{sgn}(\tilde{\beta}_{\tilde{S}}), \tag{5}$$

$$\left| \frac{1}{n}\mathbf{X}_{\tilde{S}^c}^\top \mathbf{X}_{\tilde{S}} \left(\Delta_{\tilde{S}} - w_{\tilde{S}}\right) - \frac{1}{n}\mathbf{X}_{\tilde{S}^c}^\top \varepsilon \right| \le \lambda, \tag{6}$$

*where $\tilde{S} = \{j : \tilde{\beta} \ne 0\}$ and*

$$w_j := \mathcal{S}\left(\lambda \alpha \,\mathrm{sgn}(\tilde{\beta}_j) + \Delta_j - \frac{1}{n}\mathbf{X}_j^\top \varepsilon, \lambda(1-\alpha)\right).$$

**Remark 2.** *If $\varepsilon = 0$ and $\Delta_{\tilde{S}} = 0$, the condition reduces to*

$$|\beta_{\tilde{S}}^*| > \lambda(2\alpha - 1),$$

$$\left| \frac{1}{n}\mathbf{X}_{\tilde{S}^c}^\top \mathbf{X}_{\tilde{S}} \mathcal{S}\left(\lambda \alpha \,\mathrm{sgn}(\tilde{\beta}_{\tilde{S}}), \lambda(1-\alpha)\right) \right| \le \lambda.$$

*This condition is always satisfied if $\alpha \le 1/2$.*

**Theorem 9** (Sign Unchanging Condition under Sub-Gaussian Noise). *Suppose that Assumption 1 is satisfied. Suppose that the generalized restricted eigenvalue condition (Definition 1) holds for $\mathcal{B} = \mathcal{B}(\alpha, c, \Delta)$. Assume that $\mathbf{X}_S^\top \mathbf{X}_S / n = I$. Then, we have $\mathrm{sgn}(\hat{\beta}) = \mathrm{sgn}(\tilde{\beta})$ if*

$$|\Delta_{\tilde{S}}| \le \frac{1}{2}\lambda_n,$$
$$\beta_{\min}^* > 2\lambda_n \alpha,$$
$$\left\| \frac{1}{n}\mathbf{X}_S^\top \mathbf{X}_{S^c} \right\|_\infty \le \frac{1}{(4\alpha - 1)_+}.$$

*with probability at least $1 - \exp(-n\lambda_n^2 / 8\sigma^2 + \log(2p))$.*

**Remark 3.** *This implies that the estimated sign does not change from the initial estimate if $\alpha$ and $\Delta_{\tilde{S}}$ are small enough.*

## A.3 Unchanging Condition

**Corollary 10.** *There exists a unchanging solution $\hat{\beta} = \tilde{\beta}$ if*

$$\alpha \le \frac{1}{2} \quad \text{and} \quad \max_j \left| \frac{1}{n}\mathbf{X}_j^\top r \right| \le \lambda(1 - 2\alpha).$$

*There exists a zero solution $\hat{\beta} = 0$ if*

$$\alpha \ge \frac{1}{2} \quad \text{and} \quad \max_j \left| \frac{1}{n}\mathbf{X}_j^\top \mathbf{y} \right| \le \lambda(2\alpha - 1).$$

**Remark 4.** *This is useful for constructing a search space for $\lambda$.*

# B Proofs

We give proofs as below.

## B.1 Proof of Theorem 1

*Proof.*

$$\mathcal{L}(\hat{\beta}; \tilde{\beta}) \leq \mathcal{L}(\beta^*; \tilde{\beta})$$

$$\Leftrightarrow \frac{1}{2n}\|\mathbf{X}\beta^* + \varepsilon - \mathbf{X}\hat{\beta}\|_2^2 + \lambda_n\alpha\|\hat{\beta}\|_1 + \lambda_n(1-\alpha)\|\hat{\beta} - \tilde{\beta}\|_1$$

$$\leq \frac{1}{2n}\|\varepsilon\|_2^2 + \lambda_n\alpha\|\beta^*\|_1 + \lambda_n(1-\alpha)\|\tilde{\beta} - \beta^*\|_1$$

$$\Leftrightarrow \frac{1}{2n}\|\mathbf{X}(\hat{\beta} - \beta^*)\|_2^2 + \lambda_n\alpha\|\hat{\beta}\|_1 + \lambda_n(1-\alpha)\|\hat{\beta} - \tilde{\beta}\|_1$$

$$\leq \frac{1}{n}\varepsilon^\top\mathbf{X}(\hat{\beta} - \beta^*) + \lambda_n\alpha\|\beta^*\|_1 + \lambda_n(1-\alpha)\|\tilde{\beta} - \beta^*\|_1$$

$$\Rightarrow \frac{1}{2n}\|\mathbf{X}(\hat{\beta} - \beta^*)\|_2^2 + \lambda_n\alpha\|\hat{\beta}\|_1 + \lambda_n(1-\alpha)\|\hat{\beta} - \tilde{\beta}\|_1$$

$$\leq \left\|\frac{1}{n}\mathbf{X}^\top\varepsilon\right\|_\infty \|\hat{\beta} - \beta^*\|_1 + \lambda_n\alpha\|\beta^*\|_1 + \lambda_n(1-\alpha)\|\tilde{\beta} - \beta^*\|_1$$

Because we assume that $\varepsilon$ is sub-Gaussian with $\sigma$, we have

$$\mathrm{P}\left(\left\|\frac{1}{n}\mathbf{X}^\top\varepsilon\right\|_\infty \leq \gamma_n\right) \geq 1 - \exp\left(-\frac{n\gamma_n^2}{2\sigma^2} + \log(2p)\right), \ \forall\gamma_n > 0.$$

By taking $\gamma_n = c\lambda_n$, with probability at least $1 - \exp(-nc^2\lambda_n^2/2\sigma^2 + \log(2p))$, we have

$$\frac{1}{2n}\|\mathbf{X}(\hat{\beta} - \beta^*)\|_2^2 + \lambda_n\alpha\|\hat{\beta}\|_1 + \lambda_n(1-\alpha)\|\hat{\beta} - \tilde{\beta}\|_1$$

$$\leq c\lambda_n\|\hat{\beta} - \beta^*\|_1 + \lambda_n\alpha\|\beta^*\|_1 + \lambda_n(1-\alpha)\|\tilde{\beta} - \beta^*\|_1$$

$$\Leftrightarrow \frac{1}{2n}\|\mathbf{X}(\hat{\beta} - \beta^*)\|_2^2 + \lambda_n\alpha\|\hat{\beta}_S\|_1 + \lambda_n\alpha\|\hat{\beta}_{S^c}\|_1 + \lambda_n(1-\alpha)\|\hat{\beta}_S - \tilde{\beta}_S\|_1 + \lambda_n(1-\alpha)\|\hat{\beta}_{S^c} - \tilde{\beta}_{S^c}\|_1$$

$$\leq c\lambda_n\|\hat{\beta}_S - \beta_S^*\|_1 + c\lambda_n\|\hat{\beta}_{S^c}\|_1 + \lambda_n\alpha\|\beta_S^*\|_1 + \lambda_n(1-\alpha)\|\tilde{\beta}_S - \beta_S^*\|_1 + \lambda_n(1-\alpha)\|\tilde{\beta}_{S^c}\|_1$$

$$\Rightarrow \frac{1}{2n}\|\mathbf{X}(\hat{\beta} - \beta^*)\|_2^2 + \lambda_n(\alpha - c)\|\hat{\beta}_{S^c}\|_1 + \lambda_n(1-\alpha)\|\hat{\beta}_S - \tilde{\beta}_S\|_1 + \lambda_n(1-\alpha)\|\hat{\beta}_{S^c} - \tilde{\beta}_{S^c}\|_1$$

$$\leq \lambda_n(\alpha + c)\|\hat{\beta}_S - \beta_S^*\|_1 + \lambda_n(1-\alpha)\|\tilde{\beta}_S - \beta_S^*\|_1 + \lambda_n(1-\alpha)\|\tilde{\beta}_{S^c}\|_1$$

where we used a triangular inequality $\|\beta_S^*\|_1 \leq \|\hat{\beta}_S - \beta_S^*\|_1 + \|\hat{\beta}_S\|_1$. This indicates that

$$\hat{\beta} - \beta^* \in \mathcal{B} := \{v \in \mathbb{R}^p : (\alpha - c)\|v_{S^c}\|_1 + (1-\alpha)\|v - \Delta\|_1 \leq (\alpha + c)\|v_S\|_1 + (1-\alpha)\|\Delta\|_1\}$$

where $\Delta := \tilde{\beta} - \beta^*$. On the other hand, we have

$$\frac{1}{2n}\|\mathbf{X}(\hat{\beta} - \beta^*)\|_2^2 \leq \lambda_n(\alpha + c)\|\hat{\beta}_S - \beta_S^*\|_1 + \lambda_n(1-\alpha)\|\Delta\|_1.$$

From the GRE condition, we have

$$\frac{1}{2n}\|\mathbf{X}(\hat{\beta} - \beta^*)\|_2^2 = \frac{1}{2}(\hat{\beta} - \beta^*)^\top\left(\frac{1}{n}\mathbf{X}^\top\mathbf{X}\right)(\hat{\beta} - \beta^*) \geq \frac{\phi}{2}\|\hat{\beta} - \beta^*\|_2^2.$$

Since

$$\|\hat{\beta}_S - \beta_S^*\|_1 \leq \sqrt{s}\|\hat{\beta}_S - \beta_S^*\|_2 \leq \sqrt{s}\|\hat{\beta} - \beta^*\|_2,$$

we have

$$\frac{\phi}{2}\|\hat{\beta} - \beta^*\|_2^2 \leq (\alpha + c)\lambda_n\sqrt{s}\|\hat{\beta} - \beta^*\|_2 + \lambda_n(1-\alpha)\|\Delta\|_1$$

$$\Rightarrow \|\hat{\beta} - \beta^*\|_2 \leq \frac{(\alpha + c)\lambda_n\sqrt{s} + \sqrt{(\alpha + c)^2\lambda_n^2 s + 2\phi\lambda_n(1-\alpha)\|\Delta\|_1}}{\phi}$$

$$\Rightarrow \|\hat{\beta} - \beta^*\|_2^2 \leq \frac{\left((\alpha + c)\lambda_n\sqrt{s} + \sqrt{(\alpha + c)^2\lambda_n^2 s + 2\phi\lambda_n(1-\alpha)\|\Delta\|_1}\right)^2}{\phi^2}$$

$\square$

## B.2 Proof of Theorem 2

*Proof.* From Theorem 1, we have

$$\|\hat{\beta} - \beta^*\|_2^2 \leq \frac{(\alpha + c)^2 \lambda_n^2 s}{\phi^2} \left(1 + \sqrt{1 + \frac{2(1-\alpha)\phi\|\Delta\|_1}{(\alpha+c)^2 \lambda_n s}}\right)^2$$

$$\leq \frac{(\alpha + c)^2 \lambda_n^2 s}{\phi^2} \left(2\sqrt{1 + \frac{2(1-\alpha)\phi\|\Delta\|_1}{(\alpha+c)^2 \lambda_n s}}\right)^2$$

$$= \frac{4 (\alpha + c)^2 \lambda_n^2 s}{\phi^2} + \frac{8(1-\alpha)\lambda_n\|\Delta\|_1}{\phi}$$

$$= O\left((\alpha + c)^2 \lambda_n^2 s + (1-\alpha)\lambda_n\|\Delta\|_1\right).$$

$\square$

## B.3 Proof of Theorem 3

Combining Theorem 1 and the beta-min condition conclude the assertion.

## B.4 Proof of Theorem 4 and 10

*Proof.* By KKT condition, there exists a null solution $\hat{\beta} = 0$ if and only if

$$\left|\frac{1}{n}\mathbf{X}_j^\top \mathbf{y}\right| \leq \lambda \quad \text{for} \quad \forall j \text{ s.t. } \tilde{\beta}_j = 0,$$

and

$$\left|\frac{1}{n}\mathbf{X}_j^\top \mathbf{y} + \lambda(1-\alpha)\operatorname{sgn}(\tilde{\beta}_j)\right| \leq \lambda\alpha \quad \text{for} \quad \forall j \text{ s.t. } \tilde{\beta}_j \neq 0. \tag{7}$$

(7) is equivalent to

$$-\lambda\alpha - \lambda(1-\alpha)\operatorname{sgn}(\tilde{\beta}_j) \leq \frac{1}{n}\mathbf{X}_j^\top \mathbf{y} \leq \lambda\alpha - \lambda(1-\alpha)\operatorname{sgn}(\tilde{\beta}_j) \quad \text{for} \quad \forall j \text{ s.t. } \tilde{\beta}_j \neq 0.$$

Let $r := \mathbf{y} - \mathbf{X}\tilde{\beta}$. By KKT condition, there exists an invariant solution $\hat{\beta} = \tilde{\beta}$ if and only if

$$\left|\frac{1}{n}\mathbf{X}_j^\top r\right| \leq \lambda \quad \text{for} \quad \forall j \text{ s.t. } \tilde{\beta}_j = 0,$$

and

$$\left|\frac{1}{n}\mathbf{X}_j^\top r - \lambda\alpha\operatorname{sgn}(\tilde{\beta}_j)\right| \leq \lambda(1-\alpha) \quad \text{for} \quad \forall j \text{ s.t. } \tilde{\beta}_j \neq 0. \tag{8}$$

(8) is equivalent to

$$-\lambda(1-\alpha) + \lambda\alpha\operatorname{sgn}(\tilde{\beta}_j) \leq \frac{1}{n}\mathbf{X}_j^\top r \leq \lambda(1-\alpha) + \lambda\alpha\operatorname{sgn}(\tilde{\beta}_j) \quad \text{for} \quad \forall j \text{ s.t. } \tilde{\beta}_j \neq 0.$$

$\square$

## B.5 Proof of Corollary 5

By standard theoretical analyses (or by Theorem 1 with $\alpha = 1$), we have, for some constant $c' > 0$,

$$\|\tilde{\beta} - \tilde{\beta}^*\|_2^2 \leq \frac{4(1+c')^2\lambda_m^2 s'}{\phi'^2} \quad \text{and} \quad \|\tilde{\beta} - \tilde{\beta}^*\|_1 \leq \frac{2(1+c')\lambda_m s'}{\phi'},$$

where

$$\phi' = \phi'(\mathcal{B}') := \inf_{v \in \mathcal{B}} \frac{v^\top \frac{1}{n} \mathbf{X}'^\top \mathbf{X}' v}{\|v\|_2^2} > 0, \ \mathcal{B}' = \mathcal{B}'(c') := \{v \in \mathbb{R}^p : (1-c')\|v_{S'^c}\|_1 \le (1+c')\|v_{S'}\|_1\},$$

with probability at least $1 - \nu_{m,c'}$. Hence, we have

$$\|\Delta\|_1 = \|\tilde{\beta} - \beta^*\|_1 \le \|\tilde{\beta} - \tilde{\beta}^*\|_1 + \|\tilde{\beta}^* - \beta^*\|_1 \le \frac{2(1+c')\lambda_m \tilde{s}}{\phi'} + \|\Delta^*\|_1,$$

where we define $\Delta^* := \tilde{\beta}^* - \beta^*$. Combining this and Theorem 1, we obtain the corollary.

### B.6  Proof of Theorem 6

*Proof.* By KKT condition, $\hat{\beta}$ is the estimate if and only if $\partial_\beta \mathcal{L}(\beta; \tilde{\beta}) = 0$ where $\partial$ denotes subgradient. Now, we have

$$\partial_\beta \mathcal{L}(\beta; \tilde{\beta}) = \frac{1}{n}\mathbf{X}^\top \mathbf{X}(\beta - \beta^*) - \frac{1}{n}\mathbf{X}^\top \varepsilon + \alpha\lambda\partial_\beta\|\beta\|_1 + (1-\alpha)\lambda\partial_\beta\|\beta - \tilde{\beta}\|_1.$$

where

$$\partial_{\beta_j}\|\beta\|_1 = \begin{cases} \operatorname{sgn}(\beta_j) \text{ if } \beta_j \ne 0, \\ [-1,1] \text{ if } \beta_j = 0 \end{cases} \quad \text{and } \partial_{\beta_j}\|\beta - \tilde{\beta}\|_1 = \begin{cases} \operatorname{sgn}(\beta_j - \tilde{\beta}_j) \text{ if } \beta_j \ne \tilde{\beta}_j, \\ [-1,1] \text{ if } \beta_j = \tilde{\beta}_j \end{cases}$$

Deviding $\beta = [\beta_S, \beta_{S^c}]$, we have

$$\frac{1}{n}\mathbf{X}_S^\top \mathbf{X}_S(\hat{\beta}_S - \beta_S^*) + \frac{1}{n}\mathbf{X}_S^\top \mathbf{X}_{S^c}\hat{\beta}_{S^c} - \frac{1}{n}\mathbf{X}_S^\top \varepsilon + \lambda\alpha\partial_{\hat{\beta}_S}\|\hat{\beta}_S\|_1 + \lambda(1-\alpha)\partial_{\hat{\beta}_S}\|\hat{\beta}_S - \tilde{\beta}_S\|_1 = 0$$

$$\frac{1}{n}\mathbf{X}_{S^c}^\top \mathbf{X}_S(\hat{\beta}_S - \beta_S^*) + \frac{1}{n}\mathbf{X}_{S^c}^\top \mathbf{X}_{S^c}\hat{\beta}_{S^c} - \frac{1}{n}\mathbf{X}_{S^c}^\top \varepsilon + \lambda\alpha\partial_{\hat{\beta}_{S^c}}\|\hat{\beta}_{S^c}\|_1 + \lambda(1-\alpha)\partial_{\hat{\beta}_{S^c}}\|\hat{\beta}_{S^c} - \tilde{\beta}_{S^c}\|_1 = 0$$

On the other hand, the sign consistency condition is

$$\operatorname{sgn}(\hat{\beta}_S) = \operatorname{sgn}(\beta_S^*) \text{ and } \hat{\beta}_{S^c} = 0.$$

Hence, the estimate $\hat{\beta}$ is sign consistent if and only if

$$\frac{1}{n}\mathbf{X}_S^\top \mathbf{X}_S(\hat{\beta}_S - \beta_S^*) - \frac{1}{n}\mathbf{X}_S^\top \varepsilon + \lambda\alpha\operatorname{sgn}(\beta_S^*) + \lambda(1-\alpha)\hat{w}_S = 0 \tag{9}$$

$$\frac{1}{n}\mathbf{X}_{S^c}^\top \mathbf{X}_S(\hat{\beta}_S - \beta_S^*) - \frac{1}{n}\mathbf{X}_{S^c}^\top \varepsilon + \lambda\alpha\hat{z}_{S^c} + \lambda(1-\alpha)\hat{w}_{S^c} = 0 \tag{10}$$

$$\hat{w}_j = \begin{cases} \operatorname{sgn}(\hat{\beta}_j - \tilde{\beta}_j) \text{ if } \hat{\beta}_j \ne \tilde{\beta}_j, \\ [-1,1] \text{ if } \hat{\beta}_j = \tilde{\beta}_j \end{cases} \tag{11}$$

$$|\hat{z}_{S^c}| \le 1 \tag{12}$$

$$\operatorname{sgn}(\hat{\beta}_S) = \operatorname{sgn}(\beta^*) \tag{13}$$

$$\hat{\beta}_{S^c} = 0 \tag{14}$$

Assume $\mathbf{X}_S$ is orthogonal, i.e., $\mathbf{X}_S^\top \mathbf{X}_S/n = I$. Then, from (9),

$$\hat{\beta}_S - \tilde{\beta}_S + \Delta_S - \frac{1}{n}\mathbf{X}_S^\top \varepsilon + \lambda\alpha\operatorname{sgn}(\beta_S^*) + \lambda(1-\alpha)\hat{w}_S = 0 \tag{15}$$

where $\Delta_S := \hat{\beta}_S - \tilde{\beta}_S$.

We consider the condition (15) for (i) $\hat{\beta}_j \ne \tilde{\beta}_j$ and (ii) $\hat{\beta}_j = \tilde{\beta}_j$.

(i) For $j \in S$ such that $\hat{\beta}_j \ne \tilde{\beta}_j$, from (15) and (11), we have

$$\hat{\beta}_j - \tilde{\beta}_j + \Delta_j - \frac{1}{n}\mathbf{X}_j^\top \varepsilon + \lambda\alpha\operatorname{sgn}(\beta_j^*) + \lambda(1-\alpha)\operatorname{sgn}(\hat{\beta}_j - \tilde{\beta}_j) = 0$$

$$\Rightarrow \hat{\beta}_j = \tilde{\beta}_j + \mathcal{S}\left(\frac{1}{n}\mathbf{X}_j^\top \varepsilon - \lambda\alpha\operatorname{sgn}(\beta_j^*) - \Delta_j, \lambda(1-\alpha)\right),$$

where we define the thresholding function $\mathcal{S}(u, \gamma)$ as

$$\mathcal{S}(u, \gamma) := \operatorname{sgn}(u)(|u| - \gamma)_+ = \begin{cases} u - \gamma & \text{if } u > 0 \text{ and } \gamma < |u|, \\ u + \gamma & \text{if } u < 0 \text{ and } \gamma < |u|, \\ 0 & \text{if } |u| \le \gamma. \end{cases}$$

We note that $\hat{\beta}_j \ne \tilde{\beta}_j$ requires

$$\left| \frac{1}{n} \mathbf{X}_j^\top \varepsilon - \alpha\lambda \operatorname{sgn}(\beta_j^*) - \Delta_j \right| > \lambda(1 - \alpha).$$

(ii) For $j \in S$ such that $\hat{\beta}_j = \tilde{\beta}_j$, from (15), we have

$$\hat{\beta}_j - \beta_j^* + \Delta_j - \frac{1}{n} \mathbf{X}_j^\top \varepsilon + \lambda\alpha \operatorname{sgn}(\beta_j^*) + \lambda(1 - \alpha)\hat{w}_j = 0.$$

From (11), we have

$$\left| \frac{1}{n} \mathbf{X}_j^\top \varepsilon - \lambda\alpha \operatorname{sgn}(\beta_j^*) - \Delta_j \right| \le \lambda(1 - \alpha).$$

Combining (i) and (ii), the conditions (15) and (11) reduces to

$$\hat{\beta}_S = \tilde{\beta}_S + \mathcal{S}\left( \frac{1}{n} \mathbf{X}_S^\top \varepsilon - \lambda\alpha \operatorname{sgn}(\beta_S^*) - \Delta_S, \lambda(1 - \alpha) \right). \tag{16}$$

Next, we consider the condition (10) for (i) $\hat{\beta}_j \ne \tilde{\beta}_j$ and (ii) $\hat{\beta}_j = \tilde{\beta}_j$.

(i) For $j \in S^c$ such that $\hat{\beta}_j \ne \tilde{\beta}_j$, from (10) and (14), we have

$$\frac{1}{n} \mathbf{X}_j^\top \mathbf{X}_S(\hat{\beta}_S - \beta_S^*) - \frac{1}{n} \mathbf{X}_j^\top \varepsilon + \lambda\alpha \hat{z}_j - \lambda(1 - \alpha) \operatorname{sgn}(\tilde{\beta}_j) = 0$$

By (12), we have

$$\left| \frac{1}{n} \mathbf{X}_j^\top \mathbf{X}_S(\hat{\beta}_S - \beta_S^*) - \frac{1}{n} \mathbf{X}_j^\top \varepsilon - \lambda(1 - \alpha) \operatorname{sgn}(\tilde{\beta}_j) \right| \le \lambda\alpha.$$

We note that $\hat{\beta}_j \ne \tilde{\beta}_j$ requires $\tilde{\beta}_j \ne 0$.

(ii) For $j \in S^c$ such that $\hat{\beta}_j = \tilde{\beta}_j$, from (10), (12), and (14), we have

$$\left| \frac{1}{n} \mathbf{X}_j^\top \mathbf{X}_S(\hat{\beta}_S - \beta_S^*) - \frac{1}{n} \mathbf{X}_j^\top \varepsilon \right| \le \lambda.$$

We note that $\hat{\beta}_j = \tilde{\beta}_j$ requires $\tilde{\beta}_j = 0$.

Combining (i) and (ii), the conditions (10), (12), and (14) reduces to

$$\left| \frac{1}{n} \mathbf{X}_{S^c}^\top \mathbf{X}_S(\hat{\beta}_S - \beta_S^*) - \frac{1}{n} \mathbf{X}_{S^c}^\top \varepsilon - (1 - \alpha)\lambda \operatorname{sgn}(\tilde{\beta}_{S^c}) \right| \le \lambda \left( \alpha + (1 - \alpha)\mathbb{1}\{\tilde{\beta}_{S^c} = 0\} \right). \tag{17}$$

Hence, (13), (16), and (17) concludes

$$\operatorname{sgn}\left( \beta_S^* - w_S \right) = \operatorname{sgn}(\beta_S^*),$$

$$\left| \frac{1}{n} \mathbf{X}_{S^c}^\top \mathbf{X}_S w_S + \frac{1}{n} \mathbf{X}_{S^c}^\top \varepsilon + \lambda(1 - \alpha) \operatorname{sgn}(\tilde{\beta}_{S^c}) \right| \le \lambda \left( \alpha + (1 - \alpha)\mathbb{1}\{\tilde{\beta}_{S^c} = 0\} \right),$$

where for $j \in S$,

$$w_j := -\Delta_j + \mathcal{S}\left( \lambda\alpha \operatorname{sgn}(\beta_j^*) + \Delta_j - \frac{1}{n} \mathbf{X}_j^\top \varepsilon, \lambda(1 - \alpha) \right)$$

$$= \begin{cases} \lambda(2\alpha - 1)\operatorname{sgn}(\beta_j^*) - \frac{1}{n}\mathbf{X}_j^\top\varepsilon, & \text{for } \beta_j^* > 0 \text{ and } \frac{1}{n}\mathbf{X}_j^\top\varepsilon - \Delta_j < \lambda(2\alpha - 1), \\ & \text{for } \beta_j^* < 0 \text{ and } \frac{1}{n}\mathbf{X}_j^\top\varepsilon - \Delta_j > -\lambda(2\alpha - 1), \\ -\Delta_j, & \text{for } \beta_j^* > 0 \text{ and } \lambda(2\alpha - 1) \le \frac{1}{n}\mathbf{X}_j^\top\varepsilon - \Delta_j \le \lambda, \\ & \text{for } \beta_j^* < 0 \text{ and } -\lambda \le \frac{1}{n}\mathbf{X}_j^\top\Delta_j \le -\lambda(2\alpha - 1) - \varepsilon, \\ \lambda\operatorname{sgn}(\beta_j^*) - \frac{1}{n}\mathbf{X}_j^\top\varepsilon, & \text{for } \beta_j^* > 0 \text{ and } \frac{1}{n}\mathbf{X}_j^\top\varepsilon - \Delta_j > \lambda, \\ & \text{for } \beta_j^* < 0 \text{ and } \frac{1}{n}\mathbf{X}_j^\top\varepsilon - \Delta_j < -\lambda. \end{cases}$$

$\square$

### B.7 Proof of Theorem 7

*Proof.* We derive a sufficient condition for (3) and (4).

It is sufficient for (3) that

$$\beta^*_{\min} := \min_{j \in S} |\beta^*_j| > \max_{j \in S} |w_j|.$$

Assume $|\Delta_j| \le c\lambda_n$, $0 \le c \le 1$. Because we assume that $\varepsilon$ is sub-Gaussian with $\sigma$, we have

$$\mathrm{P}\left(\left\|\frac{1}{n}\mathbf{X}_S^\top \varepsilon\right\|_\infty \le t\right) \ge 1 - \exp\left(-\frac{nt^2}{2\sigma^2} + \log(2s)\right).$$

Taking $t = (1-c)\lambda_n$, we have

$$\max_{j \in S}\left|\frac{1}{n}\mathbf{X}_j^\top \varepsilon\right| \le (1-c)\lambda_n,$$

with probability at least $1 - \exp(-n(1-c)^2\lambda_n^2/2\sigma^2 + \log(2s))$. Now, we have

$$\frac{1}{n}\mathbf{X}_j^\top \varepsilon - \Delta_j \le \left|\frac{1}{n}\mathbf{X}_j^\top \varepsilon\right| + |\Delta_j| \le (1-c)\lambda_n + c\lambda_n = \lambda_n,$$

and thus

$$\begin{aligned}
\max_{j \in S} |w_j| &\le \max_{j \in S}\left\{|2\alpha - 1|\lambda + \left|\frac{1}{n}\mathbf{X}_j^\top \varepsilon\right|, |\Delta_j|\right\} \\
&\le \max_{j \in S}\left\{|2\alpha - 1|\lambda + (1-c)\lambda_n, c\lambda_n\right\} \\
&= \lambda_n \max\left\{2 - 2\alpha - c, 2\alpha - c, c\right\}
\end{aligned}$$

Hence, $|\Delta_j| \le c\lambda_n$, $0 \le c \le 1$, and $\beta^*_{\min} > \lambda_n \max\{2 - 2\alpha - c, 2\alpha - c, c\}$ imply the condition (3).

On the other hand, it is sufficient for (4) that for $\forall j \in S^c$,

$$\left\|\frac{1}{n}\mathbf{X}_S^\top \mathbf{X}_j\right\|_\infty |w_S| + \left|\frac{1}{n}\mathbf{X}_j^\top \varepsilon\right| + \lambda_n(1-\alpha)\,\mathrm{sgn}(\tilde{\beta}_j) \le \lambda_n\left(\alpha + (1-\alpha)\mathbf{1}\{\tilde{\beta}_j = 0\}\right).$$

Since $\varepsilon$ is sub-Gaussian, we have

$$\max_{j \in S^c}\left|\frac{1}{n}\mathbf{X}_j^\top \varepsilon\right| \le (1-c)\lambda_n,$$

with probability at least $1 - \exp(-n(1-c)^2\lambda_n^2/2\sigma^2 + \log(2(p-s)))$, and

$$\max_{j \in S} |w_j| \le \lambda_n \max\left\{2 - 2\alpha - c, 2\alpha - c, c\right\}.$$

Hence, the condition (4) requires

$$\left\|\frac{1}{n}\mathbf{X}_S^\top \mathbf{X}_j\right\|_\infty \max\left\{2 - 2\alpha - c, 2\alpha - c, c\right\} + (1-c) + (1-\alpha)\,\mathrm{sgn}(\tilde{\beta}_j) \le \alpha + (1-\alpha)\mathbf{1}\{\tilde{\beta}_j = 0\},$$

that is,

$$\left\|\frac{1}{n}\mathbf{X}_S^\top \mathbf{X}_j\right\|_\infty \le \frac{c - 2(1-\alpha)\left|\mathrm{sgn}(\tilde{\beta}_j)\right|}{\max\left\{2 - 2\alpha - c, 2\alpha - c, c\right\}}.$$

(a) If $c = 1/2$, then we have a sufficient condition:

$$|\Delta_j| \le \frac{1}{2}\lambda_n,$$

$$\beta^*_{\min} > \lambda_n \max\left\{\frac{3}{2} - 2\alpha, 2\alpha - \frac{1}{2}\right\},$$

$$\tilde{\beta}_{S^c} = 0,$$

$$\left\|\frac{1}{n}\mathbf{X}_S^\top \mathbf{X}_{S^c}\right\|_\infty \le 1.$$

If $\alpha = 1/2$, then $\beta^*_{\min} > \lambda_n/2$. Under this condition, the solution has correct sign with probability at least $1 - \exp(-n\lambda_n^2/8\sigma^2 + \log(2s)) - \exp(-n\lambda_n^2/8\sigma^2 + \log(2(p-s)))$. On the other hand, if $\alpha = 1$, then it requires $\beta^*_{\min} > 3\lambda_n/2$ with the same probability.

$\square$

## B.8 Proof of Theorem 8

*Proof.* In this proof, we write $S = \{j : \tilde{\beta}_j \neq 0\}$ instead of $S = \{j : \beta_j^* \neq 0\}$. The sign invariant condition is

$$\mathrm{sgn}(\hat{\beta}_S) = \mathrm{sgn}(\tilde{\beta}_S) \text{ and } \hat{\beta}_{S^c} = 0.$$

Hence, the estimate $\hat{\beta}$ is sign invariant if and only if

$$\frac{1}{n}\mathbf{X}_S^\top \mathbf{X}_S(\hat{\beta}_S - \beta_S^*) - \frac{1}{n}\mathbf{X}_S^\top \varepsilon + \lambda\alpha\,\mathrm{sgn}(\tilde{\beta}_S) + \lambda(1-\alpha)\hat{w}_S = 0 \tag{18}$$

$$\frac{1}{n}\mathbf{X}_{S^c}^\top \mathbf{X}_S(\hat{\beta}_S - \beta_S^*) - \frac{1}{n}\mathbf{X}_{S^c}^\top \varepsilon + \lambda\alpha\hat{z}_{S^c} + \lambda(1-\alpha)\hat{w}_{S^c} = 0 \tag{19}$$

$$\hat{w}_j = \begin{cases} \mathrm{sgn}(\hat{\beta}_j - \tilde{\beta}_j) \text{ if } \hat{\beta}_j \neq \tilde{\beta}_j, \\ [-1,1] \text{ if } \hat{\beta}_j = \tilde{\beta}_j \end{cases} \tag{20}$$

$$|\hat{z}_{S^c}| \leq 1 \tag{21}$$

$$\mathrm{sgn}(\hat{\beta}_S) = \mathrm{sgn}(\tilde{\beta}) \tag{22}$$

$$\hat{\beta}_{S^c} = \tilde{\beta}_{S^c} = 0 \tag{23}$$

Assume $\mathbf{X}_S$ is orthogonal, i.e., $\mathbf{X}_S^\top \mathbf{X}_S/n = I$. Then, from (18),

$$\hat{\beta}_S - \tilde{\beta}_S + \Delta_S - \frac{1}{n}\mathbf{X}_S^\top \varepsilon + \lambda\alpha\,\mathrm{sgn}(\tilde{\beta}_S) + \lambda(1-\alpha)\hat{w}_S = 0 \tag{24}$$

where $\Delta_S := \hat{\beta}_S - \tilde{\beta}_S$.

We consider the condition (24) for (i) $\hat{\beta}_j \neq \tilde{\beta}_j$ and (ii) $\hat{\beta}_j = \tilde{\beta}_j$.

(i) For $j \in S$ such that $\hat{\beta}_j \neq \tilde{\beta}_j$, from (24) and (20), we have

$$\hat{\beta}_j - \tilde{\beta}_j + \Delta_j - \frac{1}{n}\mathbf{X}_j^\top \varepsilon + \lambda\alpha\,\mathrm{sgn}(\tilde{\beta}_S) + \lambda(1-\alpha)\,\mathrm{sgn}(\hat{\beta}_j - \tilde{\beta}_j) = 0$$

$$\Rightarrow \hat{\beta}_j = \tilde{\beta}_j + \mathcal{S}\left(\frac{1}{n}\mathbf{X}_j^\top \varepsilon - \lambda\alpha\,\mathrm{sgn}(\tilde{\beta}_j) - \Delta_j, \lambda(1-\alpha)\right).$$

We note that $\hat{\beta}_j \neq \tilde{\beta}_j$ requires

$$\left|\frac{1}{n}\mathbf{X}_j^\top \varepsilon - \alpha\lambda\,\mathrm{sgn}(\tilde{\beta}_j) - \Delta_j\right| > \lambda(1-\alpha).$$

(ii) For $j \in S$ such that $\hat{\beta}_j = \tilde{\beta}_j$, from (24), we have

$$\hat{\beta}_j - \beta_j^* + \Delta_j - \frac{1}{n}\mathbf{X}_j^\top \varepsilon + \lambda\alpha\,\mathrm{sgn}(\tilde{\beta}_j) + \lambda(1-\alpha)\hat{w}_j = 0.$$

From (20), we have

$$\left|\frac{1}{n}\mathbf{X}_j^\top \varepsilon - \lambda\alpha\,\mathrm{sgn}(\tilde{\beta}_j) - \Delta_j\right| \leq \lambda(1-\alpha).$$

Combining (i) and (ii), the conditions (24) and (20) reduces to

$$\hat{\beta}_S = \tilde{\beta}_S + \mathcal{S}\left(\frac{1}{n}\mathbf{X}_S^\top \varepsilon - \lambda\alpha\,\mathrm{sgn}(\tilde{\beta}_S) - \Delta_S, \lambda(1-\alpha)\right). \tag{25}$$

Next, we consider the condition (19). For $j \in S^c$, from (19), (21), and (23), we have

$$\left|\frac{1}{n}\mathbf{X}_j^\top \mathbf{X}_S(\hat{\beta}_S - \beta_S^*) - \frac{1}{n}\mathbf{X}_j^\top \varepsilon\right| \leq \lambda,$$

which is equivalent with

$$\left|\frac{1}{n}\mathbf{X}_{S^c}^\top \mathbf{X}_S(\hat{\beta}_S - \beta_S^*) - \frac{1}{n}\mathbf{X}_{S^c}^\top \varepsilon\right| \leq \lambda. \tag{26}$$

Hence, (22), (25), and (26) concludes

$$\text{sgn}\left(\tilde{\beta}_S + \mathcal{S}\left(\frac{1}{n}\mathbf{X}_S^\top\varepsilon - \lambda\alpha\,\text{sgn}(\tilde{\beta}_S) - \Delta_S, \lambda(1-\alpha)\right)\right) = \text{sgn}(\tilde{\beta}_S),$$

$$\left|\frac{1}{n}\mathbf{X}_{S^c}^\top\mathbf{X}_S\left(\Delta_S + \mathcal{S}\left(\frac{1}{n}\mathbf{X}_S^\top\varepsilon - \lambda\alpha\,\text{sgn}(\tilde{\beta}_S) - \Delta_S, \lambda(1-\alpha)\right)\right) - \frac{1}{n}\mathbf{X}_{S^c}^\top\varepsilon\right| \le \lambda.$$

$\square$

## B.9 Proof of Theorem 9

*Proof.* We derive a sufficient condition for (5) and (6).

It is sufficient for (5) that

$$\beta_{\min}^* := \min_{j\in S}|\beta_j^*| > \max_{j\in S}|w_j|.$$

Assume $|\Delta_j| \le c_1\lambda_n$. Because we assume that $\varepsilon$ is sub-Gaussian with $\sigma$, we have

$$\text{P}\left(\left\|\frac{1}{n}\mathbf{X}_S^\top\varepsilon\right\|_\infty \le t\right) \ge 1 - \exp\left(-\frac{nt^2}{2\sigma^2} + \log(2s)\right).$$

Taking $t = c_2\lambda_n$, we have

$$\max_{j\in S}\left|\frac{1}{n}\mathbf{X}_j^\top\varepsilon\right| \le c_2\lambda_n,$$

with probability at least $1 - \exp(-nc_2^2\lambda_n^2/2\sigma^2 + \log(2s))$. Now, we have

$$|w_j| = \left(\lambda\alpha\,\text{sgn}(\tilde{\beta}_j) + \Delta_j - \frac{1}{n}\mathbf{X}_j^\top\varepsilon - \lambda(1-\alpha)\right)_+$$

$$\le \lambda(2\alpha - 1) + |\Delta_j| + \left|\frac{1}{n}\mathbf{X}_j^\top\varepsilon\right|$$

$$\le (2\alpha + c_1 + c_2 - 1)\lambda_n.$$

Hence, $|\Delta_j| \le c_1\lambda_n$, $0 \le c \le 1$, and $\beta_{\min}^* > (2\alpha + c_1 + c_2 - 1)\lambda_n$ imply the condition (5).

On the other hand, it is sufficient for (6) that for $\forall j \in S^c$,

$$\left\|\frac{1}{n}\mathbf{X}_S^\top\mathbf{X}_j\right\|_\infty \|\Delta_S - w_S\|_\infty + \left|\frac{1}{n}\mathbf{X}_j^\top\varepsilon\right| \le \lambda_n.$$

Since $\varepsilon$ is sub-Gaussian, we have

$$\max_{j\in S^c}\left|\frac{1}{n}\mathbf{X}_j^\top\varepsilon\right| \le c_2\lambda_n,$$

with probability at least $1 - \exp(-nc_2^2\lambda_n^2/2\sigma^2 + \log(2(p-s)))$, and

$$\|\Delta_S - w_S\|_\infty \le \max_{j\in S^c}\left|\Delta_j - \mathcal{S}\left(\lambda\alpha\,\text{sgn}(\tilde{\beta}_j) + \Delta_j - \frac{1}{n}\mathbf{X}_j^\top\varepsilon, \lambda(1-\alpha)\right)\right|$$

$$\le ((2\alpha + c_2 - 1)\lambda_n)_+$$

Hence, the condition (4) requires

$$\left\|\frac{1}{n}\mathbf{X}_S^\top\mathbf{X}_j\right\|_\infty \le \frac{(1-c_2)}{(2\alpha + c_2 - 1)_+}.$$

If $c_1 = c_2 = 1/2$, we have a sufficient condition:

$$|\Delta_{\tilde{S}}| \le \frac{1}{2}\lambda_n,$$

$$\beta_{\min}^* > 2\lambda_n\alpha,$$

$$\left\|\frac{1}{n}\mathbf{X}_S^\top\mathbf{X}_{S^c}\right\|_\infty \le \frac{1}{(4\alpha - 1)_+}.$$

$\square$

Figure 6: Estimation errors of transfer learning simulations under various source and target sample sizes ($(n_s, n_t) = (500, 20), (500, 100), (100, 50)$, and $(1000, 50)$ from left to right).

## C  Additional Empirical Results

### C.1  Transfer Learning Simulation under various sample sizes

We assessed the influence of sample size of source and target domains on simulations in Section 4.2. Figure 6 shows that Transfer Lasso was more effective than others under various sample sizes.

### C.2  Newsgroup Message Data with Gradual Concept Drift

We present gradual concept drift experiments in this supplementary material, in addition to the abrupt concept drift experiments in our paper.

We used the newsgroup data[3]. In this experiment, we did not use preprocessed data[4], but instead preprocessed data in a standard manner[5]. Specifically, we first removed email headers, signatures, nested text representing quotes from other users, and stop-words. We then extracted words that totally appear more than 100 times in the whole documents and extracted documents that include at least one above word. We obtained 11066 examples and 1370 boolean bag-of-words features with 20 news topics. The topics include "comp.graphics", "comp.os.ms-windows.misc", "comp.sys.ibm.pc.hardware", "comp.sys.mac.hardware", "comp.windows.x", "rec.autos", "rec.motorcycles", "rec.sport.baseball", "rec.sport.hockey", "sci.crypt", "sci.electronics", "sci.med", "sci.space", "misc.forsale", "talk.politics.misc", "talk.politics.guns", "talk.politics.mideast", "talk.religion.misc", "alt.atheism", and "soc.religion.christian".

The objective of this problem is to predict either the user is interested in email messages or not. We suppose that the user's interest changes gradually. We randomly splitted 20 batches. The user is interested in the $k$-th through $(k + 9)$-th topics at $(2k - 1)$-th and $2k$-th batches for $k = 1, \ldots, 10$. Thus, the user's interest was stationary from $(2k - 1)$ to $2k$-th step and was gradually driftted from $2k$ to $(2k + 1)$-th step. We trained models using each batch and tested the next batch.

Figure 7: AUC (left) and coefficients (right) for newsgroup message data with gradual concept drift.

Figure 7 shows the results. Transfer Lasso outperformed others for almost all steps including stationary steps (even numbers) and gradual concept drift steps (odd numbers). Moreover, Transfer Lasso showed stable behaviors of the estimates, and some coefficients remained unchanged due to our regularization.