[Reviews · NeurIPS 2020]

Review 1

Summary and Contributions: This paper proposed an extension of Lasso named as Transfer Lasso to improve prediction under changing environment, via incorporating two L1 regularization terms, one on the (present) model parameter, and another on the differences between "source" (previous) and "target" (present) model parameters.

Strengths: 1. The problem is well motivated and the idea is clearly presented. Figures 1 and 2 help illustrating the ideas well. 2. The theoretical analyses are strong, with proofs on error bound, convergence, etc. 3. The paper is well written and easy to follow.

Weaknesses: 1. While the presented model can be posed as a transfer learning problem. This paper is more about concept drift. Therefore, the title Transfer learning via l1 Regularization is a bit too broad and can be misleading for some readers. 2. In the experiments on concept drift and "transfer learning", only Lasso is the only method studied and compared. However, Lasso is not considered as a state-of-the-art (SOTA) for concept drift and transfer learning. Lasso is designed for neither of these problem. On the other hand, there are many other methods for concept drift and transfer learning, with some discussed in the Related Work section but none is compared against in the experiment. 3. There are many existing works on concept drift (e.g. twitter activities, anomaly detection), as the authors have cited. However, this paper studies only synthetic concept drift problems. It is not clear whether the proposed solution can deal with concept drift in real data successfully. 4. Similar to the above, there are many transfer learning benchmarks and methods. However, this paper studies only a synthetic example without comparing to any transfer learning methods (as said above, Lasso is not designed for transfer learning). 5. The end of Sec. 4.2 states that Transfer Lasso showed the best accuracy in feature screening. However, previous works on Lasso screening are not cited or compared, e.g. Ren et al. "Safe feature screening for generalized LASSO." TPAMI 40.12 (2017): 2992-3006. 6. Section 4.3 follows the experiments in [17]. However, the presented results did not include [17] (and related works on the same data) in comparison. 7. Line 253: how was the data divided into 30 batches? 8. Line 258: What is the cause of such computational instability for binary features? What are the ways to mitigate this problem? 9. Figure 5-right: annotations on the colours used are missing. 10. Minor issues. Typo. Line 179: unchaing

Correctness: Some claims are misleading to me (see the above on weaknesses).

Clarity: Yes, it was quite enjoyable to read, particularly the first three sections.

Relation to Prior Work: Not really. Lasso is the primary prior work considered in this paper. Many related works are not discussed or compared.

Reproducibility: Yes

Additional Feedback:


Review 2

Summary and Contributions: In this paper, authors address the knowledge transfer via minimizing an empirical risk minimization with L1 regularization. Specifically, the method incorporates the L1 regularization of differences between source parameters and target parameters into the ordinary Lasso regularization, which ensures the sparsity of the parameters and limits the complexity of the model. This method can transfer knowledge from the past to the present in a stable environment. When the environment is unstable, it can discard the outdated knowledge and learn new knowledge, which realizes the adaptation of the model to the environment.

Strengths: The authors proposed and analyzed the L1 regularization-based transfer learning framework. This applies to any parametric models, including GLM, GAM, and deep learning.

Weaknesses: 1. The solution of the model is expressed unclearly in the initial coefficient estimation. 2. In the process of using the soft threshold function to solve the listed models, the value of the target coefficient is not given at the same time, and therefore, how are the values of the target coefficient other than the required \beta_j fixed? In addition, whether j here starts from 0 or arbitrarily specified, how to obtain the next j to be updated? 3. The sequence of cross-validation and soft threshold function for \beta is not clear. 4. The experiment is not convictive.

Correctness: The idea of the model proposed in the article is reasonable. The solution is somewhat vague. The empirical methodology is correct.

Clarity: The paper format and expression are rigorous, the theorem is proved in detail, and the algorithm part of the model solution is a bit vague.

Relation to Prior Work: Before presenting the method in the paper, the article introduces the traditional method based on hypothesis testing to detect the concept drift problem and its limitations, and then introduces the currently proposed new tree-based and holistic-based methods to achieve empirical risk through L2 regularization Minimization, but these methods are not sparse, and small changes in parameters will cause changes in the model results. Later, the proposed transfer Lasso is presented.

Reproducibility: Yes

Additional Feedback: It is recommended that the author list the steps for solving the algorithm, which will make the order of the steps clearer.


Review 3

Summary and Contributions: This paper proves theoretical results for transfer learning in a high-dimensional linear regression setup with fixed design, using a Lasso-type penalization. More specifically, given an initial estimator \tilde\beta of the unknown parameter \beta^*, the updated estimator \hat\beta minimizes a Lasso-type loss function, that forces sparsity both of \hat\beta and of \tilde\beta-\hat\beta. An extra parameter \alpha (in addition to the usual Lasso tuning parameter \lambda) allows to balance those two penalties. The error of the new estimate \hat\beta is bounded as a function of the error of the previous estimate \tilde\beta, under sub-Gaussianity of the noise and a generalized Restricted Eigenvalue condition for the fixed design. When there is no transfer (which corresponds to \alpha=1), the error seems to be similar to that of classical Lasso.

Strengths: The paper is particularly well written, with a very clear exposition of the problem and of the theoretical results, as well as the empirical evaluations. The proofs are fairly simple, since they mostly follow the standard proofs of the Lasso estimator. Moreover (although I am not at all a specialist in this field), the use of a L1 penalty for transfer learning seems new in this paper. In my opinion, this paper is absolutely relevant to the NeurIPS community.

Weaknesses: I only have one major comment, about the GRE assumption: This assumption, in Theorem 1, actually depends on the error of the initial estimate (since the set \mathcal B depends on \Detla), and hence, it seems pretty artificial to me. Could the authors elaborate a bit? For instance, if the entries of the matrix X are iid Gaussian and independent of the initial estimate, does the Assumption hold with high probability? A few minor comments: 1. In theorem 2, could you be a little more precise with the asymptotics (what quantities grow to infinity among n, p, s, and how?) 2. Line 164, replace \alpha with n in the subscript of \nu. 3. Line 179, there is a typo in the word "unchanging" 4. Right after Theorem 4, could you elaborate a little more about why it is useful?

Correctness: The claims seem all correct to me.

Clarity: The paper is very well written. Maybe it could be useful if a little more details (or arguments) were given in the proofs (such as mentioning Holder's inequality when it is used, etc.)

Relation to Prior Work: It does seem clearly discussed, although, again, I am not 100% familiar with this literature (neither Lasso nor transfer learning).

Reproducibility: Yes

Additional Feedback:


Review 4

Summary and Contributions: ---- After acknowledgement of the authors' rebuttal ---- After discussing with the fellow reviewers and considering the authors' rebuttal, I am now convinced that this submission is a worthwhile contribution. My remaining concerns are the following. 1. Motivation: the paper would be much better off if the authors gave stronger motivations for their model. Otherwise, it feels like, "Transfer Lasso is what we know how to do and analyze, therefore it will certainly have some applications somewhere"... 2. Applicability of the theoretical results: I am no longer so familiar with these types of assumptions. The authors could make an attempt to clarify when these assumptions are likely to hold, and when not. -------- The papers proposes a transfer learning approach between a source and a target domain of equal dimensions. The method is based on the Lasso estimator augmented by an $ell_1$ penalty term that encourages the target parameters to be close to the source ones. Theoretical and empirical results are provided.

Strengths: The approach is straightforward and leverages previous theoretical results for the lasso. The optimization procedure is also simple and scalable.

Weaknesses: The approach is seemingly too simple and not well-motivated: Why should the source and target parameters be close in $ell_1$ norm? The experimental results are too limited: - there is no assessment of how the sample size affects performance - no state-of-the art algorithms are being compared to, despite being mentioned throughout the paper No comparisons with transfer learning approaches that learn a common dictionary among the tasks, e.g., Maurer et al, Sparse coding for multitask and transfer learning, ICML 2013 and others.

Correctness: The optimization procedure is correct. The theoretical findings appear reasonable, however all the proofs are in the appendix, and honestly I did not check them line by line. The empirical methodology is insufficient: no impact of sample size is evaluated, and no other state-of-the-art transfer learning approaches have been considered.

Clarity: The paper is clear and well written. Personally, I would have preferred more discussions of the results, both theoretical and empirical.

Relation to Prior Work: While some previous literature has been properly discussed, many approaches to transfer learning have been ignored, most notably the ones relying on learning shared dictionaries among tasks. Furthermore, in the experimental section, the authors compare only against the Lasso, either using only the current task's training set or all the training sets combined.

Reproducibility: No

Additional Feedback: I would suggest the authors to first find stronger motivations for their approach and then perform a more exhaustive empirical comparison, especially on real datasets.

[Author Response · NeurIPS 2020]

It is a great honor for us to take your kind and meaningful feedback. We write some comments below.

**Major Comments.** We note our thinking about our experiments. Several reviewers pointed out that our experiments are insufficient. It is certainly interesting to see other comparative experiments. In this study, however, the main purpose is to propose a "simple" and "general" methodology with a link to transfer learning and concept drift. For example, the Group Lasso, Fused Lasso, Clustered Lasso, and so on, are in the same line. They have some concrete target problems, but these ideas can be easily applied to various problems. They were not the SOTA when they were proposed. This is natural because they offered general methodologies. At present, they are quite popular, because they are simple and easily applied to various problems. Our proposed method is motivated by transfer learning and concept drift, and the idea can be applied to various problems.

For the above reason, we restricted our experiments as follows. i) We focus on a high-dimensional regression problem and restrict our attention to a sparse regression model. Although this setting is important in many areas, it does not seem the mainstream of transfer learning or concept drift, so there exists limited related work. ii) We further excluded methods of transferring latent spaces, internal representations, or dictionaries in our experiments. These methods assume that source and target domains share some low-dimensional representations. This assumption is quite different from our assumption that source and target domains share model parameters (regression coefficients). We would emphasize that our method is superior to these methods in terms of interpretability and operability for continuously updating applications.

**Reviewer #1.** About weaknesses 2-4 and 6: Please see above. 5: The objective of Lasso screening such as safe screening is not accuracy but speed. These kinds of techniques can be applied to our method, but this is out of scope for this paper. 7: The data were divided into 30 batches without changing the order of the samples. 8: In the case of binary features, there are likely to exist variable pairs with very high correlations (or exactly the same variables). For $\alpha = 1/2$, the contour lines tend to be parallel to $b_j + b_k = const$ and the loss function remains equal under $b_j + b_k = const$ for $X_j \approx X_k$, making the global solution indefinite or unstable. 9: Each line is colored randomly for legibility and customary reasons (same as the `glmnet` package in R).

**Reviewer #2.** We use the ordinary coordinate descent algorithm. The formulation is convex and hence the solution does not depend on the initial value. The order of the variables to be optimized is arbitrary (can be even random). In our implementation, we used a warm-start technique for initialization and applied cyclic coordinate descent, but they do not affect results. Please check standard textbooks such as "The Element of Statistical Learning" (Section 3.8.4).

**Reviewer #3.** Thank you for the good point. It is not clear when the GRE condition will hold in general, but for $2\alpha - c - 1 > 0$, Theorem 1 and Corollary 1 in (Raskutti et al., 2010) imply that the GRE condition holds with high probability when $X_i$ is sampled from Gaussian distribution and its population covariance satisfies the GRE condition. Indeed, it can be easily shown by $\mathcal{B} \subset \{v : (2\alpha - c - 1)\|v_{S^c}\|_1 \leq (1 + c)\|v_S\|_1\}$.
About weakness 1: Theorem 2 supposes that $p$, $n$, and $s$ go to infinity and shows that $n > O(s \log(p))$ is required for convergence. 4: The search space of $\lambda$ for the ordinary Lasso is determined by $\lambda_{\max}$, the smallest value for which all coefficients are zero, and the smallest value for $\lambda$ as a fraction of $\lambda_{\max}$. In our method, we can determine $\lambda_{\max}$ as the smallest value for which all coefficients are zero or initial estimates using Theorem 4.

**Reviewer #4.** The reason why the source/target parameters should be close in $\ell_1$ norm is that we impose the assumption that the difference between source/target parameters is sparse. This sparsity assumption is reasonable under a high-dimensional online setting and informative parameters are not so many in a real-world environment, as is the Lasso assumption. We used the $\ell_1$ norm because it is a unique $\ell_q$ norm satisfying both sparsity and convexity.
We have assessed the influence of sample size, but we omitted the results due to space limitations. Figure 1 shows that Transfer Lasso was more effective than others in "4.2 Transfer Learning Simulation" under various sample sizes.

Figure 1: Estimation errors of transfer learning simulations (Section 4.2) under various source and target sample sizes $((n_s, n_t) = (500, 20), (500, 100), (100, 50),$ and $(1000, 50)$ from left to right).

[Meta-Review · NeurIPS 2020]

Though the original reviews were on the low side, after the discussion it was agreed that the paper should be primarily viewed as a *theory* paper, giving provable guarantees about a particular kind of transfer/concept drift in linear regression settings -- allowing *sparse* changes in features. On the other hand, it's also agreed that the paper *oversells* its impact in the introductory portions and the rhetoric should be somewhat toned down -- in particular, we ask the authors to point out that the main contribution is an algorithm with *theoretical guarantees* that isn't being proposed (at least as of the writing of the paper) as a competitive method with existing heuristics/algorithms. Hence -- the lack of comparison in the paper with state-of-the-art methods.